# In Vitro Regeneration of *Miscanthus* x *giganteus* through Indirect Organogenesis: Effect of Explant Type and Growth Regulators

**DOI:** 10.3390/plants10122799

**Published:** 2021-12-17

**Authors:** Aušra Blinstrubienė, Inga Jančauskienė, Natalija Burbulis

**Affiliations:** Department of Plant Biology and Food Sciences, Vytautas Magnus University Agriculture Academy, Donelaicio str. 58, 44248 Kaunas, Lithuania; i.jancauskiene@atf.viko.lt (I.J.); natalija.burbulis@vdu.lt (N.B.)

**Keywords:** explant type, growth regulators, indirect organogenesis, *Miscanthus* x *giganteus*

## Abstract

*Miscanthus* x *giganteus* is a spontaneous sterile hybrid therefore the creation of useful genetic diversity by conventional breeding methods is restricted. Plant regeneration through indirect organogenesis may be a useful approach to create genetic variability of this important agricultural crop. The present study aimed to evaluate the effect of the explant type and growth regulators on indirect organogenesis of *Miscanthus* x *giganteus* and to determine the ploidy level of plant regenerants by flow cytometry. On average, the highest percentage of morphogenic callus tested explants formed in the medium supplemented with 2.5 mg L^–1^ IBA + 0.1 mg L^–1^ BAP + 4.0 mg L^–1^ l-proline. The most intensive secondary differentiation of callus cells was observed in the medium supplemented with 4.0 mg L^–1^ ZEA + 1.0 mg L^–1^ NAA. The highest root formation frequency with the highest number of roots was determined in the MS nutrient medium supplemented with 0.4 mg L^–1^ IBA, where more than 95% of plant regenerants survived and were growing normally.

## 1. Introduction

The growing consumption of energy resources is one of the factors that contribute to global climate change. It is estimated that by 2025, global energy demand will increase by 50% [1]. The sustainable use of plant biomass and the harmonious development of renewable energy sources are becoming increasingly relevant and are one of the most important challenges of our time worldwide. In recent years, bioenergy has received a great deal of scientific attention, and energy derived from plants, especially perennial grasses and trees, would play an important role in mitigating global climate change [2]. The use of renewable energy sources is very important for the European bioeconomy under global climate change [3,4,5]. 

As fossil fuel stocks are being decreasing, *Miscanthus* x *giganteus* as renewable energy sources are increasingly used as an alternative to fossil fuels to produce biofuels, chemicals, and bioplastics [6,7,8,9,10,11,12]. In Europe and North America, increasing attention is being devoted to the cultivation of cellulose-rich perennial grasses, due to their ability to grow on low-productivity soils and their tolerance of moisture deficit [11].

*Miscanthus* x *giganteus* is distinguished from the other C4 plants for its ability to perform photosynthesis at low ambient temperatures [13,14]. Moreover, *Miscanthus* x *giganteus* can be used for phytoremediation to remove heavy metal contamination from the soil [15,16,17]. Intensive research on the hybrid is underway in order to adapt it to the unusual geographical latitudes and coldest climatic zones [18,19,20,21,22].

The development of genetic diversity of *Miscanthus* x *giganteus* with the goal to develop new cultivars with increasing productivity and resistance to biotic and abiotic factors is not possible by conventional breeding methods because according to Heaton et al. [23] it is a spontaneous sterile hybrid between *Miscanthus sacchariflorus* and *Miscanthus sinensis*. For this reason, biotechnological methods are needed for the creation of the genetic variability of *Miscanthus* x *giganteus*. Perera et al. [24] performed experiments with aim to induce mutations in elite *Miscanthus* x *giganteus* cultivar ‘Freedom’ by chemical mutagenesis through in vitro culture. Regeneration of hexaploids of *Miscanthus* x *giganteus* from immature inflorescences derived callus treated with the antimitotic agents colchicine and oryzalin in liquid and solid media was reported by Yu et al. [25]. Melnychuk et al. [26] used for polyploidization induction dinitroanilines as antimitotic agents. 

According to Krishna et al. [27], somaclonal variations which quite often occur during in vitro cultures, has been most successful in crops with limited genetic systems. Although indirect organogenesis from different explants of *Miscanthus* x *giganteus* has been carried out by various research groups [28,29,30,31,32,33], variation in ploidy level among regenerated plants as well as between plant regenerants and donor plants were only evaluated in Kim et al. [29] and Perera et al.’s [32] studies. 

The present study aimed to evaluate the effect of the explant type and growth regulators on the indirect organogenesis of *Miscanthus* x *giganteus* and to determine the ploidy level of plant regenerants by flow cytometry.

## 2. Results

### 2.1. The Effect of Explant Type and Growth Regulators on the Dedifferentiation of Somatic Cells

In the induction medium isolated explants of *Miscanthus* x *giganteus* formed callus at a frequency of 80.4 to 98.1%, depending on the explant type and growth regulators in the nutrient medium. The lowest frequency of callus formation for both tested explants was observed in the medium supplemented with 2.5 mg L^–1^ IBA (Figure 1). 

The highest callus formation frequency was observed when isolated explants had been cultivated in the medium supplemented with 5.0 mg L^–1^ IBA + 0.1 mg L^–1^ BAP + 4.0 mg L^–1^ l-proline. Under the influence of this medium composition, explants of shoot apexes and immature inflorescences produced callus at a frequency of 95.2 and 98.1%, respectively. 

The formed callus was divided into three types according to its morphological characteristics: morphogenic (greenish, compact, with meristematic foci), root-forming (yellowish) and non-morphogenic (brown). The morphogenic callus was transferred onto regeneration medium for the secondary differentiation induction.

### 2.2. The Effect of Explant Type and Growth Regulators on the Secondary Differentiation of Morphogenic Callus Cells

Callus developed by the tested explants in the regeneration medium formed shoots at a frequency ranging from 59.2% to 96.6%, depending on the interaction between the initial explant type and the growth regulators (Figure 2).

In the nutrient medium supplemented with 2.0 mg L^–1^ TDZ or 2.0 mg L^–1^ ZEA, shoot apexes developed callus formed shoots at a frequency of 61.8% or 59.2%, while immature inflorescences developed callus formed shoots at a frequency of 62.9% or 63.5%. The addition of auxin NAA at 0.5 mg L^–1^ promoted more intensive shoot formation compared with cytokinin alone, the differences are statistically significant. In the nutrient medium supplemented with 2.0 mg L^–1^ TDZ + 0.5 mg L^–1^ NAA or 2.0 mg L^–1^ ZEA + 0.5 mg L^–1^ NAA, shoots were formed at a statistically significantly higher frequency than in the medium supplemented with 2.0 mg L^–1^ TDZ or 2.0 mg L^–1^ ZEA. Increasing the concentration of TDZ and ZEA to 4.0 mg L^–1^ resulted in a statistically significant increase in the frequency of shoot formation from immature inflorescences developed callus, whereas shoot formation frequency from shoot apexes developed callus was not significantly affected by the increase in cytokinin concentration compared to 2.0 mg L^–1^ TDZ or 2.0 mg L^–1^ ZEA. The highest shoot formation frequency from the shoot apexes developed callus was observed in the nutrient medium supplemented with 4.0 mg L^–1^ ZEA + 1.0 mg L^–1^ NAA, while immature inflorescences developed callus formed shoots at the highest frequency in the nutrient medium supplemented with 4.0 mg L^–1^ TDZ + 1.0 mg L^–1^ NAA. Shoot apexes developed callus formed shoots at an average frequency of 71.2%, while immature inflorescences developed callus–at a frequency of 81.2%.

The evaluation of the number of shoots per explant showed that the lowest number of shoots (1.4) was formed by the shoot apexes developed callus in the nutrient medium supplemented with 2.0 mg L^–1^ ZEA, while the immature inflorescences developed callus formed the lowest number of shoots (2.5) in the medium supplemented with 2.0 mg L^–1^ TDZ (Figure 3).

The combinations of the tested cytokinins with 0.5 mg L^–1^ NAA resulted in an increase in shoot number per explant, especially from shoot apexes developed callus. The highest number of shoots per explant was obtained in the nutrient medium supplemented with 4.0 mg L^–1^ ZEA + 1.0 mg L^–1^ NAA. The shoot apexes developed callus formed an average of 3.9 shoots per explant, while immature inflorescences developed the callus–an average of 5.6 shoots per explant.

### 2.3. Induction of Rhizogenesis

In the MS medium without growth regulators, *Miscanthus* x *giganteus* shoots formed roots at a frequency of 57.72% (Figure 4). The addition of auxins NAA, IAA and IBA at 0.2 mg L^–1^ promoted more intensive root formation. Compared to the control, the frequency of root formation increased by 20.94%, 25.30% and 29.68%, respectively, the differences are statistically significant. When the concentration of the auxins tested had been increased to 0.4 mg L^–1^, root formation occurred in 82.86% to 98.75% of the shoots, depending on the type of auxin. The highest frequency of root formation (98.75%) was determined in the nutrient medium supplemented with 0.4 mg L^–1^ IBA. In the MS medium without growth regulators, one shoot formed an average of 3.33 roots. Due to the addition of the tested auxins at 0.2 mg L^–1^ in the nutrient medium, the average number of roots increased significantly, especially under the influence of IBA. Increasing the concentration of auxins to 0.4 mg L^–1^ resulted in an increase in the root number for all auxins tested. The highest number of roots (7.75) was observed in the nutrient medium supplemented with 0.4 mg L^–1^ IBA. 

In the WPM medium without growth regulators, *Miscanthus* x *giganteus* shoots formed roots at a frequency of 68.77% (Figure 5). 

The addition of 0.2 mg L^–1^ auxin to the nutrient medium resulted in an increase in the root formation frequency for all the auxins tested; however, compared with the control, only the shoots cultivated in the nutrient medium supplemented with 0.2 mg L^–1^ IBA showed a significantly higher root formation frequency. By increasing the concentration of auxins tested to 0.4 mg L^–1^, the root formation frequency increased by 18.45% (NAA), 13.12% (IAA) and 24.56% (IBA) compared to the control; the differences are statistically significant. The highest root formation frequency (93.33%) was observed in the nutrient medium supplemented with 0.4 mg L^–1^ IBA. In contrast to the MS medium, the addition of 0.2 mg L^–1^ of the auxins tested did not have a significant effect on the number of roots formed. Increasing the concentration of auxins to 0.4 mg L^–1^ resulted in a significant increase in the number of roots under the effect of IAA and IBA. The highest number of roots (6.41) was observed in the medium supplemented with 0.4 mg L^–1^ IBA. 

### 2.4. Acclimatization of Plant Regenerants 

Plant regenerants with well-developed roots were moved to the greenhouse, more than 95% of them survived and were growing normally.

### 2.5. Assessment of the Ploidy of Plant Regenerants

According to the scientific literature, the flow cytometry method enables the quantification of nuclear DNA [34] and the determination of ploidy [35]. Therefore, this method was applied to assess ploidy in donor plants (Figure 6a) and plant regenerants (Figure 6b,c).

A fluorescence intensity peak similar to donor plants was obtained for 12 of 15 evaluated plants regenerants (Figure 6b). However, three of the 15 plant regenerants showed histograms with two peaks (Figure 6c).

## 3. Discussion

The dedifferentiation of somatic tissue cells of *Miscanthus* x *giganteus* was induced from leaf segments, stem segments, apical meristems, immature inflorescences [28,29,30,31,32,33]. Auxin 2,4-D or combinations of 2,4-D and the cytokinin benzyladenine were most frequently used for callus induction. Głowacka et al. [28] found that explants of immature inflorescences grown in the nutrient medium supplemented with 5.0 mg L^−1^ 2,4-D and 0.1 mg L^−1^ BA produced callus at a frequency of 45–76.7%, whereas, in the study by Kim et al. [29], explants of immature inflorescences produced callus at the highest frequency (78%) in the nutrient medium supplemented with 13.6 μM 2,4-D and 0.44 μM BA. Under the influence of the combination 13.6 mM 2,4-D + 0.44 mM BA, explants of immature inflorescences formed callus at a frequency of 77% [32]. Our research suggests that explants of shoot apexes and immature inflorescences exhibited the most intensive callus formation in the medium supplemented with 5.0 mg L^−1^ IBA + 0.1 mg L^−1^ BAP + 4.0 mg L^−1^ l-proline. Our results are in agreement with Holme et al. [36] and Kim et al.’s [37] statement that the addition of proline to the nutrient medium induces a more intensive dedifferentiation of *Miscanthus* x *giganteus* cells. It was also found that the morphogenic callus formation frequency depends on the interaction between the explant type and the combination of growth regulators. The shoot apexes formed the highest percentage of morphogenic callus in the nutrient medium supplemented with 5.0 mg L^−1^ IBA + 0.1 mg L^−1^ BAP + 4.0 mg L^−1^ l-proline, while immature inflorescences formed the highest percentage of morphogenic callus in the nutrient medium supplemented with 2.5 mg L^−1^ IBA + 0.1 mg L^−1^ BAP + 4.0 mg L^−1^ l-proline. Explants of immature inflorescences produced on average 20.95% more morphogenic callus than shoot apexes. 

Głowacka et al. [28] found that immature inflorescences developed callus produced shoots at the highest frequency in the nutrient medium supplemented with 5.0 mg L^−1^ 2,4-D + 0.1 mg L^−1^ BA, whereas in the Gubišova et al. [38] studies, the secondary differentiation of cells of immature inflorescences developed callus was more strongly promoted by the combination of 5.0 mg L^−1^ 2,4-D + 2.0 mg L^−1^ BAP. Other research groups used combinations of 22.0 μM BA + 1.3 μM NAA [29,32]; 5.0 mg L^−1^ BA + 1.0 mg L^−1^ 2,4-D + 1.0 mg L^−1^ proline [37]; 3.0 mg L^–1^ BA + 0.2 mg L^−1^ NAA [31] or 0.5 mg L^−1^ KIN + 0.5 mg L^−1^ NAA [33] for the induction of secondary differentiation of cells of *Miscanthus* x *giganteus* callus. Our study showed that shoot apexes developed callus produced shoots at the highest frequency (85.3%) in the nutrient medium supplemented with 4.0 mg L^–1^ ZEA + 1.0 mg L^–1^ NAA. The combination 4.0 mg L^–1^ TDZ + 1.0 mg L^–1^ NAA promoted the most intensive formation of shoots from immature inflorescences developed callus. The frequency of shoot formation from immature inflorescences developed callus was on average 1.14 times higher in comparison with shoot apexes developed callus. A combination of growth regulators 4.0 mg L^–1^ ZEA + 1.0 mg L^–1^ NAA resulted in the highest number of shoots per explant from shoot apexes and from immature inflorescences developed callus. 

Zhao et al. [31] reported that rhizogenesis of *Miscanthus* x *giganteus* can be induced in the MS medium without growth regulators, while other research groups for rhizogenesis induction used medium supplemented with NAA [24,38,39]. Our results shown that shoots cultivated in the MS and WPM media without growth regulators formed roots at a frequency of 57.72% and 68.77%, respectively. The addition of NAA had a positive effect on root formation but was less effective than IAA and IBA supplements in the MS nutrient medium. In the WPM medium, 0.4 mg L^−1^ NAA resulted in root formation at a higher frequency compared to 0.4 mg L^−1^ IAA, but at a lower frequency compared to 0.4 mg L^−1^ IBA. The root formation frequency in the tested media was most strongly promoted by the addition of IBA, independent of the concentration. This auxin is also widely used for in vitro induction of rhizogenesis in other plants [39,40,41,42]. In our study, the highest root formation frequency (98.75%) and the highest number of roots (7.75) were observed in the MS nutrient medium supplemented with 0.4 mg L^–1^ IBA.

Perera et al. [32], in their in vitro studies on plants of the *Miscanthus* x *giganteus* cultivar ‘Freedom’, found that the somatic tissues of this cultivar form regenerants identical to the donor plant by both direct and indirect organogenesis, and therefore, both methods can be used for micropropagation. Similar results were reported by Kim et al. [29]. Our research showed that cytometric histograms of some plant regenerants were not similar to cytometric histograms of donor plants, possibly due to somaclonal variability. Future field research is needed for the evaluation of the agronomic performance of these variable plants.

## 4. Materials and Methods

### 4.1. Plant Material

Explants–shoot apexes and immature inflorescences–were collected from greenhouse-grown *Miscanthus* x *giganteus* donor plants.

### 4.2. Induction of Somatic Cells Dediferentiation

Explants were washed thoroughly under running tap water for 5 min. The surfaces of explants were disinfected in 70% ethanol for 3 min and then in 5.0% active chlorine solution supplemented with 0.05% Twin20 for 20 min. They were then washed three times with sterile distilled water and were placed on basal Murashige and Skoog (MS) [43] medium with different combinations and concentrations of indolyl-3-butyric acid (IBA), 0.1 mg L^–1^ of 6-benzylaminopurine (BAP), and 4.0 mg L^–1^ of l-proline (Table 1). 

Culture media were supplemented with 30 g L^–1^ sucrose (Sigma-Aldrich, Merck KGaA, Darmstadt, Germany) and 8 g L^–1^ Agar-Agar (Sigma-Aldrich, Merck KGaA, Darmstadt, Germany). The pH of the media was 5.7 ± 0.1. Culture media (20 mL) were dispensed into Petri dishes 90 mm in diameter and then sealed with parafilm. Nine explants were placed per Petri dish, and each treatment consisted of 12 dishes. Explants were cultivated in a growth chamber at 25/22 °C (day/night) under a 16:8 h photoperiod at a light intensity of 50 µmol m^–2^ s^–1^. The frequency of callus formation (%) was determined after 8 weeks.

### 4.3. Induction of Secondary Differentiation of Morphogenic Callus Cells

The morphogenic callus formed by shoot apexes and immature inflorescences was transferred under aseptic conditions onto regeneration medium containing different combinations and concentrations of thidiazuron (TDZ), of zeatin (ZEA), and of α-naphthylacetic acid (NAA) (Table 2). 

The nutrient media were supplemented with 30 g L^–1^ sucrose (Sigma-Aldrich, Merck KGaA, Darmstadt, Germany) and 8 g L^–1^ Agar-Agar (Sigma-Aldrich, Merck KGaA, Darmstadt, Germany). The pH of the media was 5.7 ± 0.1. The sterile culture was grown under controlled conditions: light intensity of 50 µmol m^–2^ s^–1^., photoperiod of 16/8 h (day/night), ambient temperature of 22 ± 2 °C. The frequency of shoot formation (%) and the number of shoots per explant were determined after 4 weeks.

### 4.4. Induction of Rhizogenesis

For rooting, shoots were transferred to MS and Woody Plant Medium (WPM) [44] media without auxins or supplemented with different concentrations of auxins (0.2–0.4 mg L^–1^) NAA, (0.2–0.4 mg L^–1^) IAA, and (0.2–0.4 mg L^–1^) IBA (Table 3).

The medium was solidified with 8.0 g L^–1^ Agar-Agar (Sigma-Aldrich, Merck KGaA, Darmstadt, Germany), and the pH was adjusted to 5.7. Regenerated shoots were cultivated at 25/22 °C (day/night) under a 16:8 h photoperiod at a light intensity of 50 µmol m^–2^ s^–1^. The frequency of root formation (%) and the number of roots per explant were determined after 6 weeks.

### 4.5. Acclimatization of Regenerated Plants

Plantlets with roots were removed from the medium, washed with water to remove the medium from the roots, and planted in plastic pots with perlite and vermiculite in a 1:1 ratio. The plants were initially covered with a plastic bag and maintained in a growth chamber at 25/22 °C (day/night) under a 16:8 h photoperiod at a light intensity of 150 µmol m^–2^ s^–1^ for 7 days, and then, they were transferred to the greenhouse.

### 4.6. Determination of the Ploidy Level

The ploidy level of 15 randomly selected plants regenerants was determined by PARTEC PA flow cytometer (Partec GmbH, Germany) and compared with donor plants following the methods described by Rayburn et al. [45]. 

### 4.7. Statistical Analysis

Experiments were arranged with a complete randomization and assay was performed in triplicate. The callus formation frequency as a percentage ((number of explants with callus/total number of explants) × 100%); the shoot formation frequency as a percentage ((number of explants with adventitious shoots/total number of explants) × 100%); the average number of adventitious shoots (number of adventitious shoots/number of explants forming adventitious shoots); the roots formation frequency as a percentage ((number of shoots with roots/total number of shoots) × 100%); and the average number of roots (number of roots/number of shoots with roots) were computed using the software package TIBCO Statistica, version 10 (TIBCO Software, Palo Alto, CA, USA). The statistical difference (*p* < 0.05) among the means was analyzed by Tukey post hoc test.

## Figures and Tables

**Figure 1 plants-10-02799-f001:**
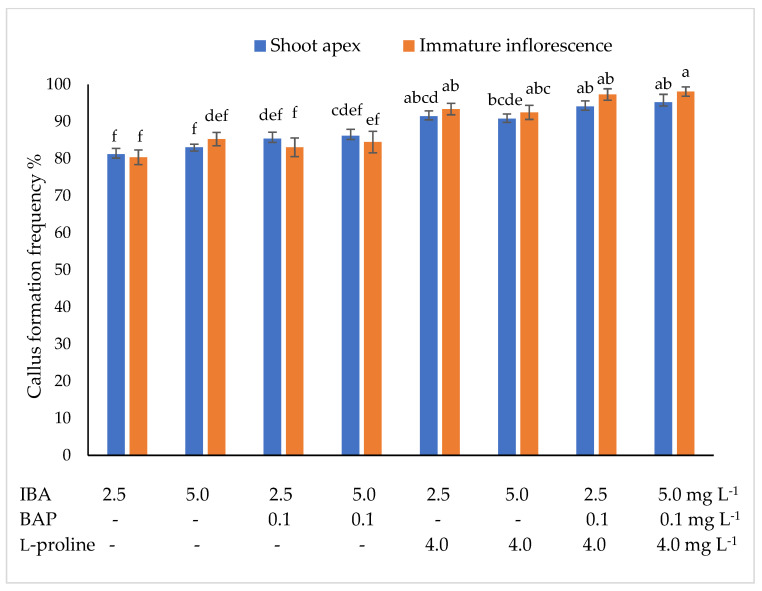
Effect of indolyl-3-butyric acid (IBA), 6-benzylaminopurine (BAP), and 4 L-proline on callus formation frequency *of Miscanthus* x *giganteus*. Treatments were performed in triplicate; statistical significance was assayed by two-way ANOVA followed by a Tukey post hoc test (*p* < 0.05). Data are expressed as mean ± standard error from 36 replicates.

**Figure 2 plants-10-02799-f002:**
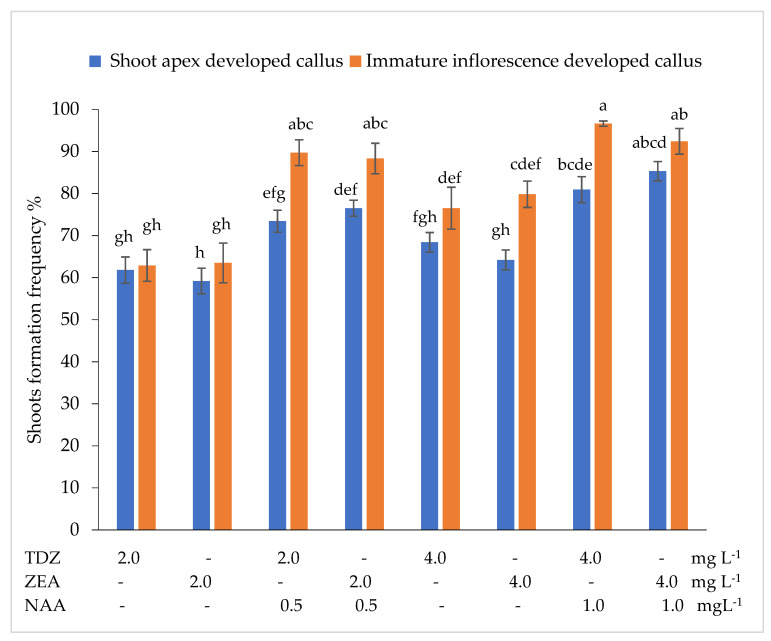
Effect of the thidiazuron (TDZ), zeatin (ZEA), and α-naphthylacetic acid (NAA) on shoots formation frequency from shoot apexes and immature inflorescences developed callus. Treatments were performed in triplicate; statistical significance was assayed by two-way ANOVA followed by a Tukey post hoc test (*p* < 0.05). Data are expressed as mean ± standard error from 36 replicates.

**Figure 3 plants-10-02799-f003:**
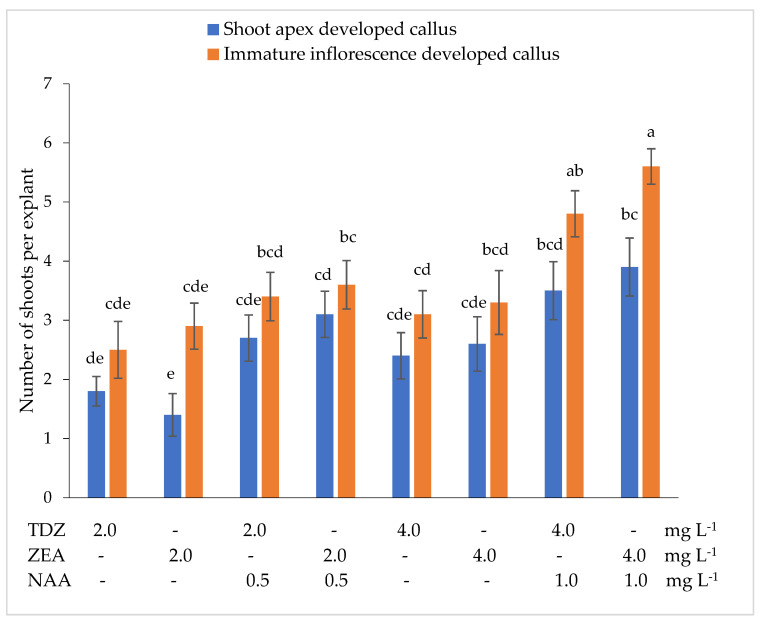
Effect of the thidiazuron (TDZ), zeatin (ZEA), and α-naphthylacetic acid (NAA) on number of shoots from shoot apexes and immature inflorescences developed callus. Treatments were performed in triplicate; statistical significance was assayed by two-way ANOVA followed by a Tukey post hoc test (*p* < 0.05). Data are expressed as mean ± standard error from 36 replicates.

**Figure 4 plants-10-02799-f004:**
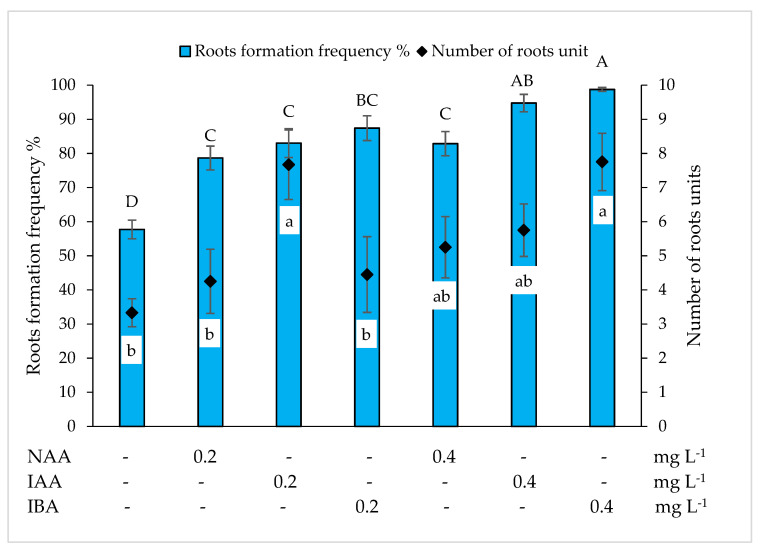
Effect of α-naphthyl acetic acid (NAA), indole-3-acetic acid (IAA) and indolyl butyric acid (IBA) on root formation frequency and number of roots *Miscanthus* x *giganteus* on MS nutrition medium. Treatments were performed in triplicate; statistical significance was assayed by one-way ANOVA followed by a Tukey post hoc test (*p* < 0.05). Data are expressed as mean ± standard error from 36 replicates.

**Figure 5 plants-10-02799-f005:**
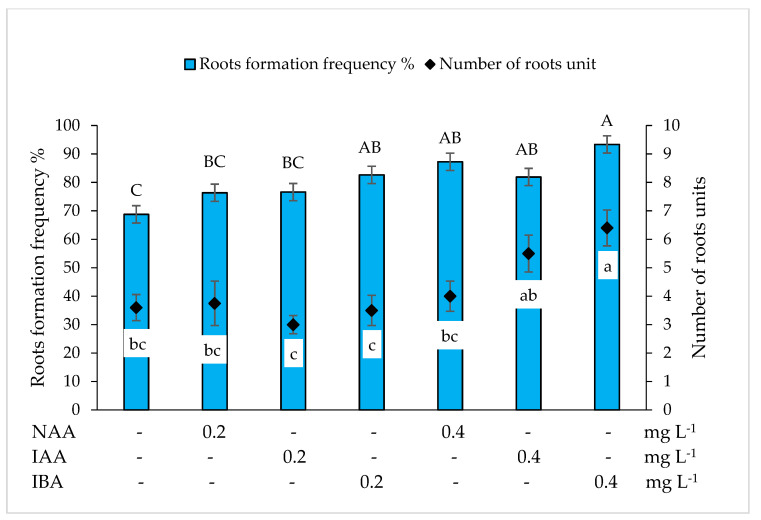
Effect of α-naphthyl acetic acid (NAA), indole-3-acetic acid (IAA) and indolyl butyric acid (IBA) on root formation frequency and number of roots *Miscanthus* x *giganteus* on WPM nutrition medium. Treatments were performed in triplicate; statistical significance was assayed by one-way ANOVA followed by a Tukey post hoc test (*p* < 0.05). Data are expressed as mean ± standard error from 36 replicates.

**Figure 6 plants-10-02799-f006:**
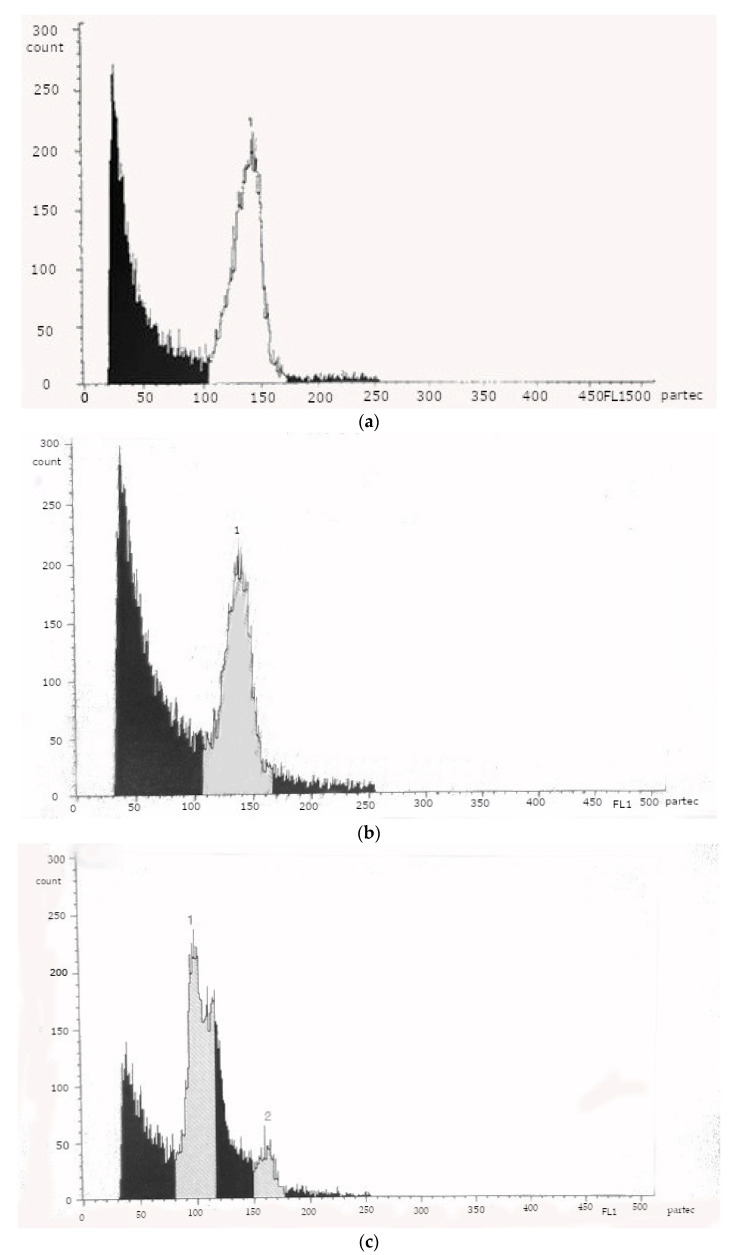
Flow cytometric histograms of donor plants (**a**) and plant regenerants (**b**,**c**).

**Table 1 plants-10-02799-t001:** Medium composition for callus induction.

Medium Number	Growth Regulator mg L^–1^		l-Proline mg L^–1^
	IBA	BAP	
I	2.5	-	-
II	5.0	-	-
III	2.5	0.1	-
IV	5.0	0.1	-
V	2.5	-	4.0
VI	5.0	-	4.0
VII	2.5	0.1	4.0
VIII	5.0	0.1	4.0

**Table 2 plants-10-02799-t002:** Medium composition for secondary differentiation induction.

Medium Number	Growth Regulator mg L^–1^		
	TDZ	ZEA	NAA
I	2.0	-	-
II	-	2.0	-
III	2.0	-	0.5
IV	-	2.0	0.5
V	4.0	-	-
VI	-	4.0	-
VII	4.0	-	1.0
VIII	-	4.0	1.0

**Table 3 plants-10-02799-t003:** Medium composition for rhyzogenesis induction.

Medium Number	MS			WPM		
	NAA mg L^–1^	IAA mg L^–1^	IBA mg L^–1^	NAA mg L^–1^	IAA mg L^–1^	IBA mg L^–1^
I	-	-	-	-	-	-
II	0.2	-	-	0.2	-	-
III	-	0.2	-	-	0.2	-
IV	-	-	0.2	-	-	0.2
V	0.4	-	-	0.4	-	-
VI	-	0.4	-	-	0.4	-
VII	-	-	0.4	-	-	0.4

## Data Availability

Not applicable.

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
