# Peer review of "In Vitro Regeneration of Miscanthus x giganteus through Indirect Organogenesis: Effect of Explant Type and Growth Regulators"

_plants, 2021, doi:10.3390/plants10122799_

Round 1

Reviewer 1 Report

The paper „In vitro Regeneration of Miscanthus x giganteus Through Indirect Organogenesis: Effect of Explant Type and Growth Regulators“ is focused on development of regeneration protocol by evaluation of the effect of explant type and different combinations of growth regulators on indirect organogenesis in Miscanthus x giganteus and on determination of ploidy level of regenerated plants by flow cytometry.

The main contribution of this paper is that it represents the study of the way how to achieve increasing in genetic diversity of Miscanthus x giganteus with goal to develop new cultivars with higher productivity and resistance to biotic and abiotic factors. Creation of genetic variability in Miscanthus x giganteus is not possible by conventional breeding methods because Miscanthus x giganteus is a spontaneous hybrid between Miscanthus sacchariflorus and Miscanthus sinensis which is sterile and for this reason biotechnological approaches must be used.

The paper is well organized and written. Some comments regarding material and methods and mainly discussion are written in the attached manuscript text. The paper can be published after minor revision.

Author Response

Dear Reviewer,

Thank you very much for so comprehensive review and very valuable suggestions. We corrected manuscript as follow:

English was corrected according yours and MDPI English Editing expert suggestions.

Point 1: Sentence “Formation of root-forming callus was most strongly stimulated by 5.0 mg L–1 IBA and combinations of 2.5 mg L–1 IBA + 0.1 mg L–1 BAP and 5.0 mg L–1 IBA + 0.1 mg L–1 BAP” required correction.

Response 1: Sentence corrected “Formation of root-forming callus was most strongly stimulated by 5.0 mg L–1 IBA and combinations of 5.0 mg L–1 IBA + 0.1 mg L–1 BAP and 5.0 mg L–1 IBA + 0.1 mg L–1 BAP”.

Point 2: The numbers in sentence “The shoot apexes developed callus formed an average of 2.7 shoots per explant, while immature inflorescences developed the callus – an average of 3.7 shoots per explant” are wrong.

Response 2: The numbers in this sentence corrected “The shoot apexes developed callus formed an average of 3.9 shoots per explant, while immature inflorescences developed the callus – an average of 5.6 shoots per explant”.

Point 3: Give table with concrete plant growth combination. The same for all experiments.

Response 3: Tables 1, 2 and 3 are included in the subsections 4.2., 4.3., and 4.4.

Table 1. Medium composition for callus induction.

Medium number

Growth regulator mg L–1

L-proline mg L–1

IBA

BAP

I

2.5

-

-

II

5.0

-

-

III

2.5

0.1

-

IV

5.0

0.1

-

V

2.5

-

4.0

VI

5.0

-

4.0

VII

2.5

0.1

4.0

VIII

5.0

0.1

4.0

Table 2. Medium composition for secondary differentiation induction.

Medium number

Growth regulator mg L–1

TDZ

ZEA

NAA

I

2.0

-

-

II

-

2.0

-

III

2.0

-

0.5

IV

-

2.0

0.5

V

4.0

-

-

VI

-

4.0

-

VII

4.0

-

1.0

VIII

-

4.0

1.0

Table 3. Medium composition for rhyzogenesis induction.

Medium number

MS

WPM

NAA mg L–1

IAA

mg L–1

IBA

mg L–1

NAA mg L–1

IAA

mg L–1

IBA

mg L–1

I

-

-

-

-

-

-

II

0.2

-

-

0.2

-

-

III

-

0.2

-

-

0.2

IV

-

-

0.2

-

-

0.2

V

0.4

-

-

0.4

-

-

VI

-

0.4

-

-

0.4

-

VII

-

-

0.4

-

-

0.4

Sincerely,

Authors

Reviewer 2 Report

I would recommend the authors to present the number of repetitions in each variant - how many explants are placed in each Petri dish, how many Petri dishes per variant.

Author Response

Dear Reviewer,

Thank you very much for carefully review and valuable suggestions.

Point 1: I would recommend the authors to present the number of repetitions in each variant - how many explants are placed in each Petri dish, how many Petri dishes per variant. 

Response 1: recommended information is provided in the subsection 4.2. “Nine explants were placed per Petri dish, and each treatment consisted of 12 dishes”.

Sincerely,

Authors

Reviewer 3 Report

Dear authors, 
I have read your manuscript very carefully and unfortunately must admit that I do not consider the work worthy of publication in this form. 
The introduction is confusing and could be much improved; 
The objective of your work is not clear and above all the data are badly represented and misinterpreted; 
The statistics should be commented on properly; 
There are no pictures although there are several references in the text to data on the colour of corns; 
English should be completely revised 

Some comments are present in the attached file 

Best regards 

Author Response

Dear Reviewer,

Thank you very much for carefully review and valuable suggestions.

Point 1: The introduction is confusing and could be much improved. 

Response 1: in the text you provided a several suggestions for the improvement and we made required corrections (please looks Response to the comments in the attached file Pages 1 and 2). But unfortunately, we don’t understand what confusing is in the introduction.

Point 2: The objective of your work is not clear and above all the data are badly represented and misinterpreted.

Response 2: the objective of our work was to evaluate the effect of the explant type and growth regulators on indirect organogenesis of Miscanthus x giganteus and to determine ploidy level of plant regenerants by flow cytometry.

Point 3: The statistics should be commented on properly.

Response 3: we stated all determined statistically significant differences among treatments through manuscript.

Point 4: There are no pictures although there are several references in the text to data on the colour of corns.

Response 4: maybe there has been some misunderstanding in yours review because there are not any references in the text to data of the colour of corns.

Point 5: English should be completely revised.

Response 5: English was corrected according to the Reviewers and MDPI English Editing expert suggestions.

Response to the comments in the attached file:

Page 1.

Number 1: useful.

We changed word viable to useful.

Number 2: a

Number 3: “…of morphogenic callus tested explants…” ??

Tested explants formed three type of callus and one of type – morphogenic callus.

Number 4: “…intensive…” what do you mean with intensive?

We mean that in this medium the highest shoot formation frequency was determined.

Number 5: the sentence “The sustainable use of plant biomass and the harmonious development of renewable energy sources are becoming increasingly relevant and are one of the most important challenges of our time worldwide” is crossed out.

We don’t understand why.

Number 6: for.

We changed word by to for.

Page 2.

Number 1: Because?? Please specify it!

As we wrote “it is sterile” and therefore cannot produce seeds.

Number 2: with co-authors

Number 3: et al.

We changed with co-authors to et al.

Number 4: of hexaploids.

We changed hexaploidy plants to hexaploids.

Number 5: Maybe it could be better if the authors review the pictures, because in this way it is not clear and not immediately informative for the reader.

How do you suggest to review the pictures?

Page 3.

Number 1: How the authors could say it?

We deleted marked sentence.

Number 2: L-proline is not a growth regulator.

We changed “Under the influence of these growth regulators…” to “Under the influence this medium composition…).

Page 4.

Figure 3. Effect of growth regulators on callus type from immature inflorescences of Miscanthus x giganteus. There is a lack of statistical analysis.

Statistical analysis provided in the Figure 3 as well as in others Figures.

Page 10.

“However, three of the fifteen plant regenerants showed histograms with two peaks”. The presence of two peaks is not the proof of polyploidization. The presence of a second peak could be escribed also to an intensive mitotic activity!

Actually, we did say that three of the fifteen plant regenerants are polyploids, we just stated what their histograms are with two peaks

Page 11. How the pH has been adjusted?

pH has been adjusted with sodium hydroxide.

Page 12. The description of Statistical analysis must be re-written.

That is wrong in description

“Experiments were arranged with a complete randomization. The callus formation frequency as a percentage ((number of explants with callus/total number of explants) x 100%); the shoot formation frequency as a percentage ((number of explants with adventitious shoots/total number of explants) x 100%), the average number of adventitious shoots (number of adventitious shoots/number of explants forming adventitious shoots); the roots formation frequency as a percentage ((number of shoots with roots/total number of shoots) x 100%) and the average number of roots (number of roots/number of shoots with roots) were computed using the software package TIBCO Statistica, version 10 (TIBCO Software, Palo Alto, CA, USA). The mean value of shoot formation frequency and corresponding SE for every treatment were computed based on the number of independent replications. Fisher’s least significant difference (LSD) test was carried out with a significance level of p < 0.01.”

What exactly you suggest us to correct?

Sincerely,

Authors
